# The impact of outpatient settlement for cross-regional medical treatment on healthcare choice by the floating population: PSM+DID evidence based on CFPS

**Qiang Su, Yumeng Zhang, Yuanhao Sui, Lihua Sun****\*, Dawei Zhang\***

Department of Pharmacy Administration, School of Business Administration, Shenyang Pharmaceutical University, Shenyang, Liaoning, China

\* slh-3632@163.com (LS); nkelite@163.com (DZ)

## Abstract

The issue of regional disparities in medical resources and insurance benefits is notably prevalent in China. As a result, seeking medical treatment across regions has emerged as a focal point of health reform efforts in recent years. This study examines the impact of outpatient settlement for cross-regional medical treatment on health-care choices by the floating population who live and work in regions different from their registered residence. Using data from the China Family Panel Studies (CFPS), this study conducts an empirical analysis through Propensity Score Matching and Difference-in-Differences (PSM + DID). The analysis included a sample of 3,019 floating population from 2012 to 2020. The findings indicate that they are significantly more inclined to select advanced medical institution (coefficient = 0.252, p < 0.05) due to the influence of the policy. Furthermore, heterogeneity analysis reveals that this policy primarily affects the labor force (coefficient = 0.268, p < 0.05) and urban resi-dents (coefficient = 0.286, p < 0.05) within the floating population. Ultimately, the out-patient settlement for cross-regional medical treatment affects the floating population to seek care at advanced medical institution, which conflicts with the objectives of the hierarchical medical system. Policymakers should enhance communication to ensure better coordination between these policies.

## Introduction

According to the latest Chinese national population census, there has been an observable increase in population mobility, accompanied by an inconsistency between household registration and actual residential locations [1]. This phenomenon has led to an increasing misalignment between individuals' health insurance enroll-ment locations and their actual places of residence [2]. In particular, with the increas-ing demand for high-quality medical resources and the concentration of inter-regional

**Data availability statement:** The original data presented in the study are openly available in the Institute of Social Sciences Survey (ISSS) of Peking University at https://www.isss.pku.edu.cn/.

**Funding:** The author(s) received no specific funding for this work.

**Competing interests:** The authors have declared that no competing interests exist.

migrants in economically developed regions [3], this misalignment results in the underutilization of healthcare resources and a decrease in the overall efficiency of medical resource allocation. Such challenges exacerbate disparities in healthcare resource distribution and social health insurance benefits inequalities. Consequently, cross-regional medical treatment has become a focal point in healthcare reform efforts [2]. As early as March 2009, the Opinions of the Central Committee of the Communist Party of China and the State Council on Deepening the Reform of the Medical and Healthcare System [4] proposed prioritizing the transfer and continuation of basic medical insurance for rural migrant workers and improving settlement services for retirees receiving medical care cross-regional. This initiative launched direct settlement services for cross-regional medical expenses, initially focusing on hospitalization costs [5]. Following the establishment of the National Healthcare Security Administration (NHSA), the scope of cross-regional medical reform has been further expanded. In May 2018, the NHSA launched cross-regional direct settlement for outpatient expenses [6]. By September of the same year, the Yangtze River Delta region (Shanghai, Jiangsu, Zhejiang, and Anhui) was selected as the first pilot region due to its high population mobility and pressing demand for cross-regional medical treatment [7]. In December 2019, the southwest region (Sichuan, Chongqing, Guizhou, Yunnan, and Xizang) and the Beijing-Tianjin-Hebei region also initiated pilot programs for outpatient expense settlement [8]. These pilots were designed to test the policy's feasibility in regions with varying economic development levels and healthcare resource distributions. On the basis of the initial success of these pilots, in September 2020, the NHSA and the Ministry of Finance jointly issued the Notice on the Issuance of the List of Newly Added Pilot Provinces for Cross-Regional Direct Settlement of Outpatient Expenses, signaling the nationwide implementation of cross-regional direct settlement for outpatient expenses [9].

The direct settlement of cross-regional medical expenses has established a unified network for cross-regional medical insurance settlements, enabling interconnectivity among different medical insurance pooling areas and overcoming the limitations of the traditional localized management system. Patients seeking medical treatment cross-regional can complete the required pre-treatment filing procedures in advance through various online and offline channels, including the National Medical Insurance Service Platform app, the mini-program on the State Council mobile for cross-regional medical services, or the service windows of insurance agencies. Once the filing is completed, patients are able to access medical services at designated medical institutions using valid credentials such as the electronic medical insurance certificate or social security card. When settling medical expenses, individuals only need to pay the portion they are personally responsible for, without the need to advance payments or go through additional procedures to receive medical insurance reimbursement [10]. This measure has significantly simplified the reimbursement process for cross-regional medical insurance patients, enhancing the convenience and efficiency of medical insurance services. However, the increased convenience may also lead to complex interactions and impacts on the hierarchical medical system.

The primary objective of the hierarchical medical system is to optimize the allocation of medical resources by promoting the division of labor and cooperation among various levels of medical institutions. This aims to address the issue of overcrowding in advanced medical institutions and underutilization of resources in primary healthcare facilities, ultimately achieving a healthcare delivery model where patients with minor illnesses are treated in community-level institutions, serious illnesses are treated in general hospitals and recovery occurs in the community. Nevertheless, the increasing accessibility of outpatient settlement for cross-regional medical treatment would motivate a part of insured individuals to seek treatment at advanced medical institutions outside their registered residence. This trend could lead to a rise in outpatient visits to these advanced medical institutions, thereby hindering the implementation of the hierarchical medical system. However, research on this phenomenon remains limited.

Cross-regional medical treatment has become prevalent in Europe and the United States. Due to differences in healthcare systems abroad compared to China, such practices are generally referred to as "medical tourism" or "cross-border healthcare" in international contexts [11,12]. International scholars primarily focus on two aspects of Cross-regional medical treatment in their research. The first aspect is the driving factors of cross-regional medical treatment. Scholars have noted that it is driven by factors such as the pursuit of life extension, distrust of regional doctors, and the demand for more convenient and higher-quality medical services [13–17]. The second aspect is the impact of cross-regional medical treatment. Findings indicate that, due to convenience and quality, patients tend to utilize more healthcare resources and prefer advanced medical institutions to primary, which ultimately results in elevated medical costs [18–20]. Compared with international research, Chinese scholars have predominantly concentrated on the benefits of outpatient settlement for cross-regional medical treatment. Studies [21–23] have demonstrated that outpatient settlement for cross-regional medical treatment enhances the convenience of flotation individuals, promoting public welfare. Additionally, Zhong Yuyin [24] analyzed settlement data for medical insurance before and after the implementation of outpatient settlement for cross-regional medical treatment in pilot cities, revealing that this policy led to an increase in patient visits to advanced medical institutions, which poses challenges to the effective implementation of the hierarchical medical system. Similarly, using hospital data, Yang Xi and her team [25] found that the average cost per visit for patients utilizing cross-regional settlement was approximately 5,000 Chinese Yuan (CNY) higher than that of locally insured patients. Existing studies primarily focus on cities and hospitals as research subjects to examine the impact of outpatient settlement for cross-regional medical treatment. However, there is a lack of research exploring the influence of this policy on individuals' healthcare choices.

Based on the data from the China Family Panel Studies (CFPS), this research empirically analyzed the impact of outpatient settlement for cross-regional medical treatment on the healthcare choice of the floating population by combining propensity score matching and difference in difference (PSM + DID). This research addresses gaps in existing studies at the individual level and provides insights for policy formulation and personalized adjustments worldwide based on China's experience by examining the varied responses of different floating individuals to this outpatient reimbursement policy.

## Materials and methods

### Data sources

This study utilizes the China Family Panel Studies (CFPS) (https://www.isss.pku.edu.cn/), a comprehensive and multidisciplinary public database established by the Institute of Social Sciences Survey (ISSS) of Peking University. The data for this research were sourced from 2012 to 2020. The data were screened based on the study subjects and the logic of the survey questionnaire, applying the following inclusion and exclusion criteria, a final sample size of 3,019 individuals was included: 1) Individuals with missing household registration information for any year were excluded; 2) Individuals whose residence and household registration information were the same at any year were excluded; 3) Only those individuals whose residence and household registration were both located within mainland China were included.

## Variables

This study assesses the impact of outpatient reimbursement for cross-regional medical treatment on the healthcare choice of floating individual. Utilizing their responses to survey questions regarding medical institution choices from the CFPS as the dependent variable, denoted as "hq". Based on the policy's pilot area and implementation period, key independent variables include "treat" (indicate whether the policy was implemented in the region), "time" (indicate whether the policy was implemented at this time), and the interaction term "did" (capturing the policy's effect).

In conducting propensity score matching, it is essential to select factors that influence the treatment variable "treat" as covariates [26,27]. Consequently, variables such as "shouru" (personal annual income), "hl" (level of the medical institution where treatment was sought), "bod" (degree of trust in doctors), and "soh" (personal health status), which affect the treatment variable "treat", are chosen as covariates for this analysis.

Previous research [28–32] has demonstrated that various factors influence an individual's healthcare choice. Therefore, drawing on prior studies and practical considerations from the CFPS dataset, this study includes these factors as control variables, such as "age", "itlg" (intelligence level), and "hpl" (hospitalization in the past year), among others. The detailed description is provided in Table 1.

## Statistical methods

In this study, PSM + DID was used to assess the impact of outpatient reimbursement for cross-regional medical treatment on floating individual's healthcare choices. Concurrently, to explore potential disparities in the policy's impact across different demographic groups, the study categorizes participants based on age, employment status, and type of residence to facilitate a heterogeneity analysis.

**Table 1. Meaning of variables.**

| Variable classification | | Label | Definition and Assignment |
|---|---|---|---|
| Dependent variable | | hq | Locations typically visited when seeking medical attention from a doctor, rated on a scale from 1 to 5: general hospital = 5, specialized hospital = 4, community health service center/township health center = 3, community healthcare station/village clinic = 2, private clinic = 1. |
| Key independent variables | | treat | The implementation of policy in each region. If the policy was implemented in this region = 1, otherwise = 0. |
| | | time | The implementation of policy each year. If the policy was implemented this year = 1, otherwise = 0. |
| | | did | The interaction term of treat and time. If the policy was implemented in the region and year = 1, otherwise = 0. |
| Control variables | | hpl | Hospitalization due to illness in the past 12 months. If there was a hospitalization within the past 12 months = 1, otherwise = 0. |
| | | chronic | Diagnosis of a chronic illness by a doctor within the past 6 months. If an individual had a chronic illness within the past 6 months = 1, otherwise = 0. |
| | | itlg | The intelligence level of an individual is rated from very low to very high across 1–7. |
| | | age | The age of the individual for this year. |
| | | lnshouru | Logarithm of Shouru. |
| | | bx | Whether the type of medical reimbursement available can be used for cross-regional outpatient reimbursement. If it can be used = 1, otherwise = 0. |
| | Covariates for PSM | hl | Rating of the medical treatment level at the medical institution, classified on a scale of 1–5, from very low to very high. |
| | | bod | The individual's level of trust in doctors. Which is measured on a scale of 0–10, ranging from completely distrustful to completely trustworthy. |
| | | soh | The health status of an individual. Which is classified on a scale of 1–5, from very poor to very good. |
| | | shouru | Individual earnings from their jobs over the past 12 months. (Unit: Chinese Yuan) |

Traditional DID analysis necessitates that policy implementation times remain consistent among individuals in the experimental group. However, the policy of outpatient reimbursement for cross-regional medical treatment is introduced on a provincial basis, resulting in varying implementation times for individuals in the experimental group. Consequently, this study utilizes multi-period DID [33] to assess the differences before and after intervention between the experimental and control groups, both of which implement the policy at different time points. Previous research [34] has demonstrated that the optimal matching effect is achieved when the number of neighbors in the propensity score matching method is four. Accordingly, this study sets the number of neighbors for matching control groups to four for the experimental group, allowing for year-by-year matching. Estimates of DID are obtained using the Ordinary Least Squares (OLS) method, and the model is structured as follows:

$$hq_{it}^{psm} = \alpha + \beta did_{it} + \delta x_{it} + u_i + \lambda_t + \varepsilon_{it}$$

(1)

In this study, "psm" denotes regression conducted on sample data that satisfies the criteria for common support following PSM. The variable labeled as $hq_{it}$ represents the dependent variable, illustrating the healthcare choice of the surveyed individual i in year t. The variable labeled as $did_{it}$ is a double difference term that signifies whether the surveyed individual i is eligible for reimbursement for outpatient medical treatment cross-regional in year t. If reimbursement is approved, its value is 1. Otherwise, it is equal to 0. $\beta$ represents the estimation coefficient, illustrating the direct impact of outpatient reimbursement for cross-regional medical treatment on the healthcare choice for floating individual. The variable labeled as $x_{it}$ represents the control variable, $u_i$ represents the individual fixed effect, $\lambda_t$ represents the time fixed effect and $\varepsilon_{it}$ represents the disturbance term. Additionally, clustering is executed at the individual level to account for the clustered standard error. All analyses were performed using Stata 17.

### Ethical approval

The CFPS is a nationally representative, biennial household survey that has been performed since March 2010, organized by the Institute of Social Science Survey, Peking University. The Peking University Biomedical Ethics Review Committee provided ethical approval for the survey (Approval number: IRB00001052–14010). All respondents read a statement that explained the purpose of the study and gave written consent to continue (with guardians providing consent for minors).

## Results

### Descriptive statistics

The descriptive statistics of the variables are presented in Table 2. The sample included 3019 individuals, with a male-to-female ratio of 51.34% (n = 1550) and 48.66% (n = 1465), an average age of 37 years, 70.12% residing in towns (n = 2117), and 29.88% in rural areas (n = 902). Regarding employment status, 79% were employed (n = 2385), 8.21% were unemployed (n = 248), 12.97% had withdrawn from the labor market (n = 386), and the mean annual individual income was 26,133.93 yuan. In terms of health status, 43.52% had reimbursable medical insurance (n = 1314), 9.97% had experienced chronic diseases within six months (n = 301), and 6.23% had been hospitalized for illness in the past 12 months (n = 188).

### Benchmark regression

Table 3 presents the coefficient estimate of the impact of outpatient reimbursement for cross-regional medical treatment on the healthcare choices of floating individual. The coefficients of the double difference term are all positive at the 5% significance level, indicating that outpatient reimbursement for cross-regional medical treatment significantly impacts the healthcare choices of floating individual, leading to a preference for advanced medical institution for medical treatment.

**Table 2. Descriptive statistics of variables.**

| Variable classification | | Label | Mean | Standard Deviation | Minimum | Maximum |
|---|---|---|---|---|---|---|
| Dependent variable | | hq | 3.531 | 1.588 | 1.000 | 5.000 |
| Key independent variables | | treat | 0.503 | 0.500 | 0.000 | 1.000 |
| | | time | 0.628 | 0.483 | 0.000 | 1.000 |
| | | did | 0.132 | 0.338 | 0.000 | 1.000 |
| Control variables | | hpl | 0.062 | 0.242 | 0.000 | 1.000 |
| | | chronic | 0.100 | 0.300 | 0.000 | 1.000 |
| | | itlg | 5.619 | 1.254 | 1.000 | 7.000 |
| | | age | 37.003 | 11.857 | 16.000 | 89.000 |
| | | lnshouru | 5.482 | 5.222 | 0.000 | 13.459 |
| | | bx | 0.435 | 0.496 | 0.000 | 1.000 |
| | Covariates for PSM | hl | 2.987 | 0.951 | 1.000 | 5.000 |
| | | bod | 6.508 | 2.290 | 0.000 | 10.000 |
| | | soh | 2.760 | 1.110 | 1.000 | 5.000 |
| | | shouru | 26133.930 | 46327.770 | 0.000 | 700000.000 |

**Table 3. Baseline regression results.**

| variable | Return (1) | Return (2) |
|---|---|---|
| did | 0.274** | 0.252** |
| hpl | | 0.457*** |
| chronic | | 0.347*** |
| itlg | | 0.088*** |
| age | | –0.062 |
| lnshouru | | 0.022** |
| hl | | –0.049 |
| bod | | 0.047** |
| bx | | 0.140 |
| soh | | –0.003 |
| Constant term | 3.347*** | 4.475** |
| Observations | 2728 | 2728 |
| $R^2$ | 0.0140 | 0.0059 |

Both regressions (1) and (2) use data with nonzero weights after propensity score matching. The treatment and control groups consist of 1,357 and 1,371. Individual fixed effects and time-fixed effects are adopted, with clustering at the individual level. Regression (1) does not include control variables, while regression (2) includes control variables. *, **, *** indicate significance levels of 10%, 5% and 1%, and this notation applies to all subsequent tables.

## Heterogeneity analysis

In this study, grouping regression was used based on individual age, type of urban-rural residency, and employment status. Additionally, a cross-regression of employment status and type of urban-rural residency was conducted to facilitate heterogeneity analysis. Given that the data source for this study exclusively includes individuals over the age of 16 who have household registration information, the age is categorized into two stages by the demographic standards set by the National Bureau of Statistics: individuals aged 16–64 are classified as the working-age population, while those aged 65 years and older are classified as the elderly population.

The findings of the heterogeneity analysis are presented in Table 4. The regression coefficient of the double difference term indicates that outpatient reimbursement for cross-regional medical treatment significantly influences the healthcare choice of the employed population, the working age cohort, and urban residents. Specifically, this policy encourages them to choose advanced medical institution for their treatments. In contrast, it does not have a significant impact on unemployed, elderly, or rural residents.

All regressions use data with non-zero weights after propensity score matching and employ individual and time-fixed effects while clustering at the individual level.

## Robustness checks

### Common support assumption

Following the application of propensity score matching, it is essential to ensure compliance with the common support assumption and conduct balance tests. As illustrated in Fig 1, the quantity of experimental and control groups within the corresponding score intervals is comparable, thus satisfying the common support assumption.

**Table 4. Heterogeneity analysis.**

| Variable | Sample | | | | | | | |
|---|---|---|---|---|---|---|---|---|
| | Town | Rural | Working age population | Elderly population | Employed people | Unemployed people | Urban employment population | Rural employment population |
| did | 0.286** | 0.095 | 0.273** | 0.302 | 0.268 ** | 0.461 | 0.298** | 0.061 |
| N | 1907 | 821 | 2633 | 95 | 2147 | 581 | 1498 | 649 |
| $R^2$ | 0.0108 | 0.0082 | 0.0124 | 0.0474 | 0.0151 | 0.0105 | 0.0131 | 0.0077 |

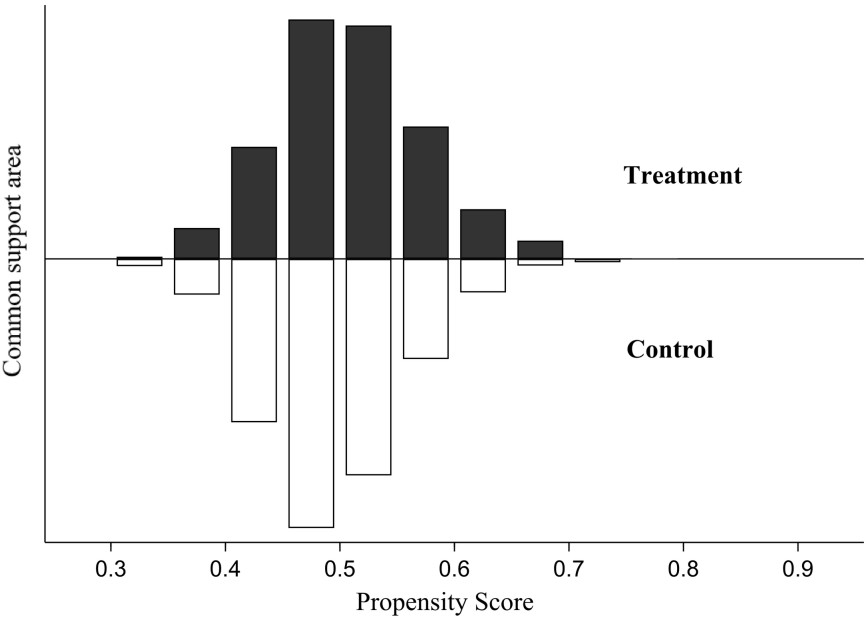

**Fig 1. Common support assumption.**

## Balance test

The t-test results for the experimental and control groups, both pre and post matching, are presented in Table 5. It is evident that, except in 2016, there was a significant disparity between the experimental and control groups regarding personal annual income (shouru) before matching each year. Additionally, a notable difference was observed between the two groups regarding the degree of trust in doctors (bod) and personal health status (soh) in 2012. However, after the matching process, no significant difference was identified between the experimental and control groups across all variables in each year.

As illustrated in Figs 2 and 3, the kernel density plots of propensity scores both before and after matching indicate that the average propensity scores of the experimental and control groups are closer after matching, as verified by balance tests. Consequently, it can be concluded that the results of the propensity score matching are satisfactory.

**Table 5. The t-test for covariates before and after matching across years.**

|  | Covariates | Sample | Year | | | | |
|---|---|---|---|---|---|---|---|
|  |  |  | 2012 | 2014 | 2016 | 2018 | 2020 |
| Regression Coefficients of Covariates | shouru | Before | 0.000** | 0.000** | −0.000 | 0.000*** | 0.000*** |
|  |  | After | 0.000* | 0.000 | −0.000 | 0.000 | 0.000 |
|  | hl | Before | −0.065 | −0.143 | −0.044 | 0.005 | 0.059 |
|  |  | After | 0.017 | −0.159 | −0.063 | 0.030 | 0.049 |
|  | bod | Before | 0.099** | −0.011 | 0.015 | 0.014 | 0.047 |
|  |  | After | 0.049 | −0.007 | 0.029 | 0.012 | 0.033 |
|  | soh | Before | 0.155** | 0.064 | 0.029 | 0.024 | 0.090 |
|  |  | After | 0.104 | 0.075 | 0.027 | 0.001 | 0.114 |
| N |  | Before | 649 | 625 | 635 | 592 | 518 |
|  |  | After | 572 | 561 | 595 | 533 | 467 |
| R² |  | Before | 0.0167 | 0.0087 | 0.0009 | 0.0120 | 0.0178 |
|  |  | After | 0.0069 | 0.0031 | 0.0019 | 0.0020 | 0.0039 |

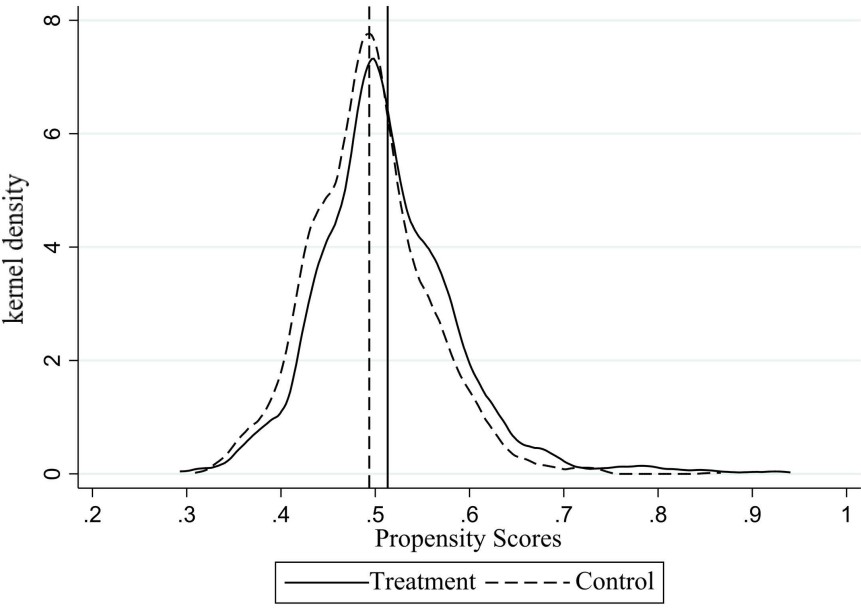

**Fig 2. Propensity score kernel density plot before the match.**

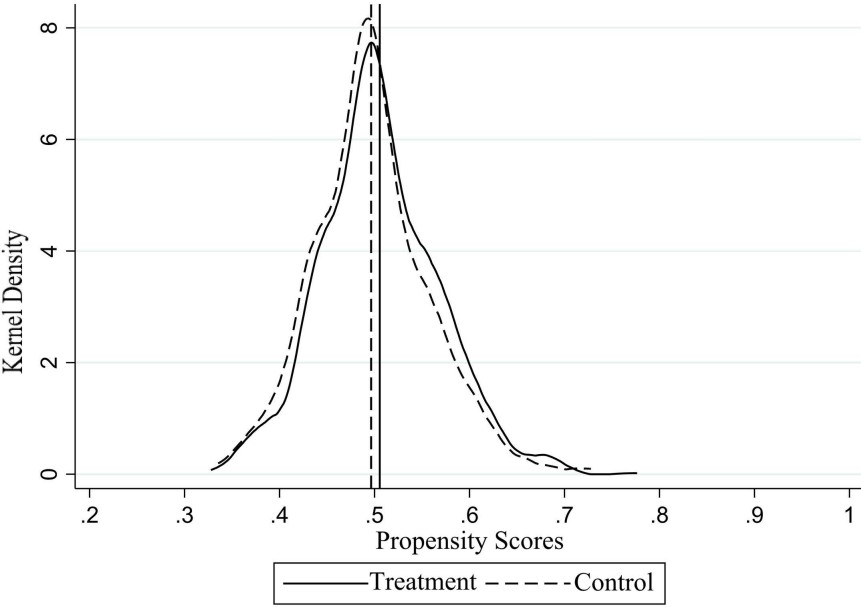

**Fig 3. Propensity score kernel density plot after the match.**

## Parallel trends test

The DID method is based on the assumption of a parallel trend, which stipulates that in the absence of policy or intervention, the trajectory of outcome variables for both experimental and control groups would remain identical. In this research, the event study method was used to integrate the terms between the dummy variables at each time point and the dummy variables representing the experimental group into the regression model for the parallel trend test. This approach facilitates an exploration of policy impact effects across varying periods. To address multicollinearity, one data period needs to be omitted, then the second period before policy implementation is eliminated. The results of this test are presented in Fig 4. The coefficients associated with the interaction terms before the policy implementation are not statistically significant, suggesting no substantial disparities between the experimental and control groups before policy implementation, thereby validating the parallel trend assumption.

## Placebo test

In this study, 500 samples were randomly selected from the total sample using the individual placebo test. For each draw, a pseudo-experimental group and a pseudo-policy implementation time are generated, along with a pseudo-did term derived from their product for baseline regression to obtain estimated coefficients. As illustrated in Fig 5, most pseudo-did regression coefficients are insignificant at the 5% level. The mean distribution of the total pseudo-did regression coefficients is around zero, which significantly deviates from the actual did term coefficient of 0.252, thereby passing the individual placebo test.

## Model substitution test

Meanwhile, the robustness of regression models was tested by fixed effect, cluster robust standard deviation and propensity score matching conditional replacement regression. The test results are presented in Table 6. Notably, the did regression coefficient of Regression (3) is significantly positive at the 10% level, both the did regression coefficients of

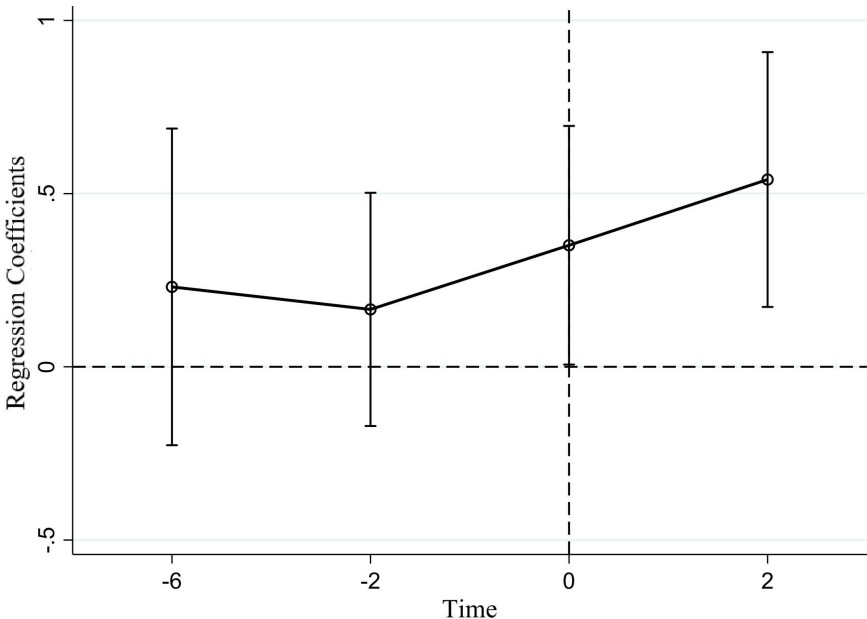

**Fig 4. Parallel trends test.**

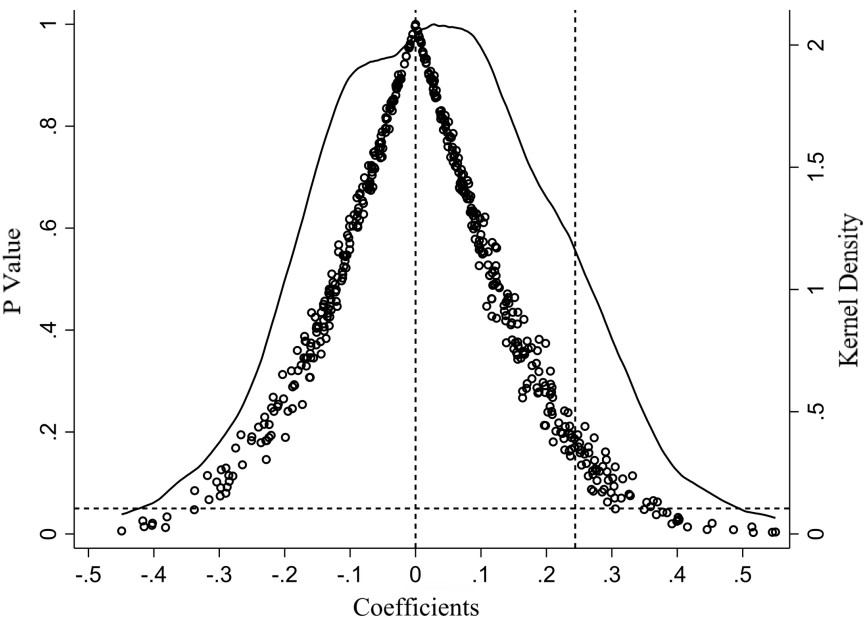

**Fig 5. Placebo test.**

Regressions (4) and (5) are significantly positive at the 5% level, and those of Regressions (1) and (2) are significantly positive at the 1% level. In summary, following conditional replacement on the regression, the did regression coefficient remains significant, indicating strong model stability.

**Table 6. Model substitution test.**

| variables | Regression(1) | Regression(2) | Regression(3) | Regression(4) | Regression(5) |
|---|---|---|---|---|---|
| did | 0.482*** | 0.466*** | 0.209* | 0.252** | 0.252** |
| hpl | 0.427*** | 0.488*** | 0.466*** | 0.457*** | 0.457*** |
| chronic | 0.414*** | 0.379*** | 0.336** | 0.347*** | 0.347*** |
| itlg | 0.110*** | 0.110*** | 0.104*** | 0.088*** | 0.088*** |
| age | −0.002 | −0.001 | 0.016 | −0.062 | −0.062 |
| lnshouru | 0.013** | 0.014** | 0.018** | 0.022** | 0.022** |
| hl | −0.090*** | −0.089*** | −0.049 | −0.049 | −0.049 |
| bod | 0.027** | 0.035*** | 0.043** | 0.047** | 0.047** |
| bx | 0.024 | 0.053 | 0.148 | 0.140 | 0.140 |
| soh | 0.021 | 0.031 | −0.003 | −0.003 | −0.003 |
| Constant term | 2.769*** | 2.662*** | 1.951*** | 4.475** | 4.475** |
| PSM | NO | YES | YES | YES | YES |
| Individual fixed effects | NO | NO | YES | YES | YES |
| Time fixed effects | NO | NO | NO | YES | YES |
| Cluster | NO | NO | NO | NO | YES |
| N | 3019 | 2728 | 2728 | 2728 | 2728 |
| $R^2$ | 0.0417 | 0.0398 | 0.0199 | 0.0059 | 0.0059 |

## Discussion

This study enhances existing research by utilizing CFPS survey data to examine the impact of China's outpatient reimbursement policy for cross-regional medical treatment on the healthcare choice of floating individual. Employing the PSM+DID method, the findings indicate that the outpatient reimbursement for cross-regional medical treatment policy encourages floating individuals to seek medical treatment from advanced medical institution, which conflicts with the goal of the hierarchical medical system, resulting in policy inconsistency.

The phenomenon can be attributed to two primary reasons. First, the existing pilot program for outpatient reimbursement for cross-regional medical treatment primarily focuses on advanced medical institution. As a result, patients are forced to visit these institutions to receive reimbursements. Second, due to policy incentives, the disproportionate distribution of medical resources induces residents in resource-deficient areas to seek medical treatment in resource-abundant regions.

To mitigate the strain on medical resources caused by uneven distribution, it is essential to improve the development of medical treatment infrastructure in underprivileged areas. Promoting collaboration between medical institution in regions with limited resources and advanced medical institution is crucial. Additionally, optimizing the utilization of existing facilities, equipment, and skilled teams will accelerate regional medical treatment standards enhancement.

Moreover, heterogeneity analysis indicates that the healthcare choices of urban residents are significantly shaped by policy, resulting in a preference for advanced medical institutions for their healthcare needs. In contrast, the healthcare preferences of rural residents remain largely unaffected. This disparity arises from the limitations inherent in the policy pilot. Urban residents and employees are more likely to access information about the policy and to select advanced medical institutions nearby, while rural residents encounter challenges due to transportation issues and information asymmetry.

To enhance policy implementation and promote equity, it is essential to gradually expand the policy coverage to encompass more grassroots medical institutions within the reimbursement system. Additionally, providing clear guidance through village committees and primary medical institutions can assist rural residents in understanding and actively engaging with the policy.

The current pilot program for outpatient reimbursement for cross-regional medical treatment primarily focuses on advanced medical institutions in towns, restricting patients' medical choices and disrupting the efficient allocation of resources within the hierarchical medical system. In policy formulation and implementation, the pharmaceutical, medical insurance, and healthcare sectors must balance public fairness and convenience with the efficient allocation of medical resources. To facilitate this, advanced medical institutions should be encouraged to support primary healthcare institutions by deploying experts to grassroots healthcare and leveraging telemedicine. These approaches would enhance the capabilities of primary healthcare institutions and foster collaborative mechanisms for resource sharing. The reimbursement process should also be refined, and ratios should be adjusted to promote organized medical treatment. This would reduce unnecessary visits to advanced medical institutions while ensuring that the outpatient reimbursement for cross-regional medical treatment policy aligns effectively with the hierarchical medical system.

Of course, there are certain limitations to this study. To facilitate readers' comprehension, the DID regression in this study was conducted using the widely adopted OLS method. Future research could consider employing various methods simultaneously, such as Generalized Linear Models (GLM) and Machine Learning (ML) methods, for DID regression. On this basis, the optimal model can be selected and could explore more comprehensible interpretations of the results for the optimal model.

## Conclusions

In this study, PSM + DID methods were employed to conduct empirical research. The findings indicate that outpatient reimbursement for cross-regional medical treatment encourages the floating population to seek medical treatment at advanced medical institution, resulting in discordance with the hierarchical medical system. This study provides empirical evidence from an individual perspective, addressing gaps in existing research on the impact of outpatient reimbursement for cross-regional medical treatment on healthcare choice. Additionally, the study offers recommendations for balancing public fairness and convenience with the efficient allocation of medical resources while ensuring alignment between the outpatient reimbursement for cross-regional medical treatment policy and the hierarchical medical system. Moreover, the findings contribute insights for enhancing such policies in countries with similar floating population and hierarchical medical system, and they also furnish pertinent information for countries that have yet to implement such policies.

## Author contributions

**Conceptualization:** Qiang Su, Lihua Sun, Dawei Zhang.

**Data curation:** Yuanhao Sui.

**Formal analysis:** Qiang Su.

**Methodology:** Qiang Su, Dawei Zhang.

**Resources:** Yumeng Zhang.

**Software:** Qiang Su.

**Validation:** Qiang Su.

**Writing – original draft:** Qiang Su, Yumeng Zhang, Yuanhao Sui.

**Writing – review & editing:** Qiang Su, Lihua Sun, Dawei Zhang.

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
