## [Decision Letter · Decision Letter 0]

13 Dec 2024

PONE-D-24-36231The Impact of Outpatient Reimbursement for Cross-Regional Medical Services on the Decision of Medical Treatment by the Floating Population: PSM-DID evidence from China's CFPSPLOS ONE

Dear Dr. Sun,

Thank you for submitting your manuscript to PLOS ONE. After careful consideration, we feel that it has merit but does not fully meet PLOS ONE’s publication criteria as it currently stands. Therefore, we invite you to submit a revised version of the manuscript that addresses the points raised during the review process. Both reviewers have provided evaluations that acknowledge the potential of the paper. However, several areas require significant improvement, particularly concerning the excessive generality of certain statements, the lack of clarity in various sections of the text, and numerous points necessitating clarification regarding methodological aspects. Furthermore, it is recommended to strengthen the final policy implications. These issues are reported comprehensively in the two reports. My recommendation to the authors is to carefully consider all the critiques and suggestions contained therein to resubmit a revised version of the manuscript that effectively addresses the observations provided.

We look forward to receiving your revised manuscript.

Kind regards,

Matteo Lippi Bruni, PhD

Academic Editor

PLOS ONE

Journal Requirements:

When submitting your revision, we need you to address these additional requirements. 1. Please ensure that your manuscript meets PLOS ONE's style requirements, including those for file naming. The PLOS ONE style templates can be found at https://journals.plos.org/plosone/s/file?id=wjVg/PLOSOne_formatting_sample_main_body.pdf and https://journals.plos.org/plosone/s/file?id=ba62/PLOSOne_formatting_sample_title_authors_affiliations.pdf 2.  Please amend either the title on the online submission form (via Edit Submission) or the title in the manuscript so that they are identical. 3. Please match your authorship list in your manuscript file and in the system. 4. Please include a caption for figure 1, 2 and 5. 5. PLOS requires an ORCID iD for the corresponding author in Editorial Manager on papers submitted after December 6th, 2016. Please ensure that you have an ORCID iD and that it is validated in Editorial Manager. To do this, go to ‘Update my Information’ (in the upper left-hand corner of the main menu), and click on the Fetch/Validate link next to the ORCID field. This will take you to the ORCID site and allow you to create a new iD or authenticate a pre-existing iD in Editorial Manager.

Reviewers' comments:

Reviewer's Responses to Questions

**Comments to the Author**

1. Is the manuscript technically sound, and do the data support the conclusions?

Reviewer #1: Yes

Reviewer #2: Partly

2. Has the statistical analysis been performed appropriately and rigorously? 

Reviewer #1: Yes

Reviewer #2: I Don't Know

3. Have the authors made all data underlying the findings in their manuscript fully available?

Reviewer #1: Yes

Reviewer #2: Yes

4. Is the manuscript presented in an intelligible fashion and written in standard English?

Reviewer #1: No

Reviewer #2: Yes

5. Review Comments to the Author

Reviewer #1: This paper investigates the impact of cross-regional outpatient reimbursement policies on the medical treatment choices of floating populations in China using a combination of Propensity Score Matching (PSM) and Difference-in-Differences (DID) methodologies. The results provide empirical evidence that these reimbursement policies significantly encourage floating individuals to seek care at higher-tier medical institutions, which may be at odds with the goals of China’s hierarchical medical system. The study uses a robust dataset (China Family Panel Studies, CFPS) and presents thorough analysis techniques.

Here’s a detailed review based on various aspects of the manuscript:

1. Clarity and language:

The manuscript is generally well-organized and follows a logical structure. However, there are a few areas where the paper could benefit from more clarity (e.g., grammar issues, conciseness) or elaboration, especially in the abstract and introduction.

For example, the phrasing of the second sentence in the abstract “This study seeks to investigate the influence of outpatient reimbursement for cross-regional medical services on floating individual medical treatment decision” could be reworded as: “This study examines how outpatient reimbursement for cross-regional medical services influences the medical treatment decisions of floating individuals.”

Again, “Empirical analysis suggest that floating population are more likely…” (line 25) should be “Empirical analysis suggests that the floating population is more likely…”. “Cross-regional outpatient reimbursement policies effect floating population” (line 29): this should be “affect” instead of “effect.”

More generally, the use of terms like floating population” and “high-tier medical institutions” are somewhat vague and could benefit from more precise definitions early in the paper to avoid ambiguity for readers less familiar with Chinese healthcare policy.

Overall, I suggest having the paper proofread by an English native speaker before further submitting it.

2. Outpatient reimbursement reform for cross-regional medical services:

The explanation of the outpatient reimbursement reform in the study could benefit from much more clarity. Specifically, they mention the introduction of a cross-regional reimbursement system for outpatient services as part of China’s healthcare reform, but there’s a lack of detailed explanation about how this policy is structured and implemented in real-world terms. For a better understanding, several aspects of the reform could have been elaborated further, such as:

- Policy Mechanism: While the study mentions that the reimbursement for cross-regional medical services was initially focused on hospitalization and later expanded to outpatient services, it doesn’t explain the process in detail. How do individuals access care in different regions, and what steps are involved in the reimbursement process? How do patients know if they are eligible for reimbursement or how to apply for it?

- Pilot Areas: The authors refer to the pilot regions where the reimbursement system was first introduced, but do not explain how these geographical areas were chosen. Do these regions have specific geographical or demographic characteristics? Are they more urban or rural? Did these characteristics play a role in the selection process?

- Coordination with Hierarchical Medical System: The authors argue that the policy conflicts with the goals of the hierarchical medical system, but they do not fully explain how the hierarchical system is supposed to work in the first place. A clearer explanation of this system, and why the cross-regional reimbursement reform disrupts it, would help readers better understand the policy's implications (see also comment below).

3. Literature review:

The literature review section highlights the significance of cross-regional medical services in China and points out that there is limited research on the effect of outpatient reimbursement policies on individual treatment choices. The references used seem relevant, but the author could expand on the findings and limitations of existing studies more explicitly, which could provide stronger justification for the present study.

4. Methods:

The methodology section is clearly articulated, with detailed explanations of the data source (CFPS) and statistical techniques (PSM+DID). The authors have justified their choice of methods, citing relevant studies. The use of PSM to account for selection bias and DID to examine policy effects is appropriate for this type of analysis.

• Sample Size: The sample size (3,019 individuals) is sufficient for the analysis, but the authors should explain how the sample was chosen more explicitly. Was there any weighting or criteria in the sample selection process? This should be clarified.

• Variable Description: The explanation of variables (e.g., medical choice, treatment, income, etc.) is mostly clear. However, “Medical choice” as the dependent variable could use a more detailed description. How exactly is this variable measured? It would help the reader to understand its operationalization in the context of this study. Furthermore, the statement “To assess the influence of the outpatient reimbursement policy...” could be rewritten for clarity. Instead of “variables such as 'treatment' (treat), 'policy time' (time), and their product double difference term (did),” you could say: “Key independent variables include 'treatment' (indicating whether the individual sought treatment), 'policy time' (indicating the implementation period of the policy), and the interaction term (did) that captures the policy’s effect.”

• Model Specification: The use of the multi-period DID method is a strength, as it accounts for varying policy implementation times across provinces. However, more detail could be provided regarding the reasoning for the specific model used and the control variables. For instance, why were certain covariates like “rust in doctors” and “health status” chosen, and how might these influence treatment decisions?

5. The Impact of the Reform on Different Groups: Although the study finds that certain groups (urban residents, employed individuals) are more affected by the reform, there’s not enough context about why this might be the case. What features of the reform make it particularly impactful for these groups? Is it due to their access to information, ability to navigate the system, or perhaps their greater reliance on higher-level medical institutions in urban settings?

6. Policy implications:

While both discussion and conclusion sections briefly touch on policy recommendations, they could delve deeper into the policy implications of the findings. Specifically, while the paper’s findings show that cross-regional reimbursement encourages higher-tier institution use, the authors do not fully address how the policy could be improved to align better with the goals of the hierarchical medical system (e.g., promoting more equitable access, ensuring efficient resource allocation). A deeper analysis of potential trade-offs and policy levers could be beneficial.

Reviewer #2: Referee Report - PONE-D-24-36231

The Impact of Outpatient Reimbursement for Cross-Regional Medical Services on the

Decision of Medical Treatment by the Floating Population: PSM-DID evidence from

China's CFPS

The authors present a paper about the impact of cross-regional outpatient reimbursement policies on the medical treatment decisions of the floating population in China. Using data from the China Family Panel Studies (CFPS) spanning 2012 to 2020, the authors adopt a Propensity Score Matching and a Difference-in-Differences methodologies for their empirical analysis.

The study highlights policy-induced shifts in healthcare-seeking behavior, revealing mismatches between policy objectives and outcomes. It suggests a need for better coordination among healthcare policies, improved infrastructure in under-resourced areas, and adjustments to ensure alignment with hierarchical system goals.

The findings offer insights for policymakers aiming to refine reimbursement policies and address healthcare inequities in China.

I have some concerns on the paper summarized in the following:

1) In the introduction the authors affirm that “According to the most recent national census data from China, there is a notable phenomenon of household separation. This has exacerbated disparities in the distribution of medical resources and insurance benefits across various. regions”. This sentence is generic, and I do not understand what it means. I think that more specific explanation is needed.

2) In the introduction some papers on cross-regional patients’ migration are discussed, but they seem focused only on China. I think that Plos One is devoted to an international panel of researchers and then the state of art and the rest of the paper should be focused to provide a contribution for a most extended audience.

3) For the data analysis China Family Panel Studies is used, but it is not clear how the authors select 3,019 individuals. They state that this is a sample, but I do not understand if it is a representative sample or not, if they applied any selection criteria etc. All this part needs to be clarified.

4) The dependet variable hq is measured on a scale between 1 to 5 (see Table 1), but the equation (1) seems to be estimated using an OLS approach. Even in this case I do not understand how the authors approach the estimation process.

5) In Table 2, where the regression coefficients are presented, N is equal to 2,728, a reduced number compared to the overall 3,019. Is this a consequence of the PSM application? In addition, the authors cite the paper number 24 and they say that “matching effect is optimal when the number of neighbors in the propensity score matching method is four”. Does this mean that in the 2,728 selected observations we have a ratio of 1:4 between treated and untreated?

6) I would like to see a Table of descriptive statistics where the main variables considered are split between treated and control units.

7) It is not clear in the conclusion the contribution of the paper to an international audience.

8) A brief explanation of the healthcare system should be included.

Minor comments:

1) Over all the text there are several points, starting from abstract and introduction, where a blank space between words is missing.

2) In page 7 Table 3 is indicated as Table 2.

6. PLOS authors have the option to publish the peer review history of their article (what does this mean? ). If published, this will include your full peer review and any attached files.

**Do you want your identity to be public for this peer review?** For information about this choice, including consent withdrawal, please see our Privacy Policy .

Reviewer #1: No

Reviewer #2: No

---

## [Author Response · Author response to Decision Letter 0]

15 Jan 2025

Date: Jan 15, 2025

Manuscript Title: The impact of outpatient settlement for cross-regional medical treatment on healthcare choice by the floating population: PSM+DID evidence based on CFPS

Manuscript ID: PONE-D-24-36231

Dear Dr. Matteo and reviewers,

We sincerely appreciate your recognition of our study titled “The impact of outpatient settlement for cross-regional medical treatment on healthcare choice by the floating population: PSM+DID evidence based on CFPS” (ID: PONE-D-24-36231) and the comments you provided. Your insights have significantly contributed to improving our manuscript, and we are deeply grateful for the time and effort you dedicated to reviewing our work. We have carefully reviewed each of your comments and have thoroughly revised the manuscript, clarifying unclear points, refining the policy background for better reader comprehension, and clarifying the contribution of our study to an international audience. Below, we provide the original comments and our responses, with our responses marked in italics for clarity. (The PDF file generated by Editorial Manager cannot display italic formatting)

Response to Dr. Matteo

1. Comment: Please ensure that your manuscript meets PLOS ONE's style requirements, including those for file naming. The PLOS ONE style templates can be found at https://journals.plos.org/plosone/s/file?id=wjVg/PLOSOne_formatting_sample_main_body.pdf and https://journals.plos.org/plosone/s/file?id=ba62/PLOSOne_formatting_sample_title_authors_affiliations.pdf

1. Response: We appreciate your careful review of the manuscript's formatting. We have revised and corrected the title capitalization, author addresses, and the font and size throughout the manuscript in accordance with the PLOS ONE’s style requirements. We hope that the new version of the manuscript is now in total accord with the Journal formatting requirements.

2. Comment: Please amend either the title on the online submission form (via Edit Submission) or the title in the manuscript so that they are identical.

2. Response: We have carefully verified both the title in the online submission and the title in the manuscript, making revisions to ensure they are identical.

3. Comment: Please match your authorship list in your manuscript file and in the system.

3. Response: Following your suggestion, we have matched the authorship list in the manuscript with the one in the system. Additionally, due to Dr. Dawei Zhang's contributions during the response to reviewers stage, he will be added as a co-corresponding author, and we have made revisions to the author order and corresponding authors information.

Original:

“Qiang Su, Yumeng Zhang, Yuanhao Sui, Dawei Zhang and Lihua Sun*

Department of Pharmacy Administration, School of Business Administration, Shenyang Pharmaceutical University, 103 Wenhua Road, Shenyang, 110016, Liaoning Province, PR China

* Corresponding author

E-mail: slh-3632@163.com(SLH)”

Revised:

“Qiang Su1, Yumeng Zhang1, Yuanhao Sui1, Lihua Sun1,*, Dawei Zhang1,*

1 Department of Pharmacy Administration, School of Business Administration, Shenyang Pharmaceutical University, Shenyang, Liaoning Province, China

* Corresponding authors

E-mail: slh-3632@163.com (LS); nkelite@163.com (DZ)”

4. Comment: Please include a caption for figure 1, 2 and 5.

4. Response: We sincerely appreciate your thorough review and apologize for any inconvenience caused. After a careful check, we found that the original manuscript already included the figure titles for figures 1, 2, and 5. In the Revised Manuscript with Track Changes, we have highlighted all figure titles in yellow (on lines 261, 275, 276, 290, and 300, respectively). The figure titles in the manuscript are as follows:

Fig 1. Common support assumption.

Fig 2. Propensity score kernel density plot before the match.

Fig 3. Propensity score kernel density plot after the match.

Fig 4. Parallel trends test.

Fig 5. Placebo test.

5. Comment: PLOS requires an ORCID iD for the corresponding author in Editorial Manager on papers submitted after December 6th, 2016. Please ensure that you have an ORCID iD and that it is validated in Editorial Manager. To do this, go to ‘Update my Information’ (in the upper left-hand corner of the main menu), and click on the Fetch/Validate link next to the ORCID field. This will take you to the ORCID site and allow you to create a new iD or authenticate a pre-existing iD in Editorial Manager.

5. Response: Thank you very much for your reminder. The corresponding author, Lihua Sun, has linked the ORCID (0009-0001-1705-6313).

Response to Reviewer #1

1. Comment: This paper investigates the impact of cross-regional outpatient reimbursement policies on the medical treatment choices of floating populations in China using a combination of Propensity Score Matching (PSM) and Difference-in-Differences (DID) methodologies. The results provide empirical evidence that these reimbursement policies significantly encourage floating individuals to seek care at higher-tier medical institutions, which may be at odds with the goals of China’s hierarchical medical system. The study uses a robust dataset (China Family Panel Studies, CFPS) and presents thorough analysis techniques.

1. Response: We sincerely appreciate the time and effort you dedicated to thoroughly reviewing our manuscript and accurately summarizing the overall content of our study. We are truly honored to have received your recognition of our manuscript.

2. Comment: Clarity and language: The manuscript is generally well-organized and follows a logical structure. However, there are a few areas where the paper could benefit from more clarity (e.g., grammar issues, conciseness) or elaboration, especially in the abstract and introduction. For example, the phrasing of the second sentence in the abstract “This study seeks to investigate the influence of outpatient reimbursement for cross-regional medical services on floating individual medical treatment decision” could be reworded as: “This study examines how outpatient reimbursement for cross-regional medical services influences the medical treatment decisions of floating individuals.” Again, “Empirical analysis suggest that floating population are more likely…” (line 25) should be “Empirical analysis suggests that the floating population is more likely…”. “Cross-regional outpatient reimbursement policy effect floating population” (line 29): this should be “affect” instead of “effect.” More generally, the use of terms like “floating population” and “high-tier medical institutions” are somewhat vague and could benefit from more precise definitions early in the paper to avoid ambiguity for readers less familiar with Chinese healthcare policy. Overall, I suggest having the paper proofread by an English native speaker before further submitting it.

2. Response: We are grateful for your recognition of our manuscript's structure and logical consistency, as well as for your thoughtful and detailed revision suggestions. We have thoroughly addressed the grammatical issues and made improvements to the areas where the expressions were not concise or clear:

1) To ensure that readers unfamiliar with Chinese healthcare policies can more easily understand the content, we have refined the language and thoroughly reviewed the entire manuscript. We replaced “high-tier medical institutions” with “advanced medical institution” and added an explanation the first time “floating population” appears in the manuscript, specifying that it refers to a population “…who live and work in regions different from their registered residence.” (on lines 24-25 of the manuscript and lines 27-28 of the revised manuscript with track changes.)

2) According to your suggestion, the phrasing of the second sentence in the abstract, “This study seeks to investigate the influence of outpatient reimbursement for cross-regional medical services on floating individual medical treatment decision” has been reworded as: “This study examines the impact of outpatient settlement for cross-regional medical treatment on healthcare choices by the floating population...” (on lines 23-24 of the manuscript and lines 25-27 of the revised manuscript with track changes.)

3) The sixth sentence of the abstract, “Empirical analysis suggest that floating population are more likely…” has been reworded as: “The findings indicate that they are significantly more inclined to select…”(on lines 27-28 of the manuscript and lines 32-33 of the revised manuscript with track changes.)

4) The “effect” in the seventh sentence of the abstract has been revised to “affect”. (on line 29 of the manuscript and line 35 of the revised manuscript with track changes.)

5) It is a great suggestion to have an English native speaker proofread the manuscript before submission. The manuscript has been proofread by an English native speaker, and the revisions have been marked in the file labeled 'Revised Manuscript with Track Changes' using the Word revision mode.

3. Comment: Outpatient reimbursement reform for cross-regional medical services: The explanation of the outpatient reimbursement reform in the study could benefit from much more clarity. Specifically, they mention the introduction of a cross-regional reimbursement system for outpatient services as part of China’s healthcare reform, but there’s a lack of detailed explanation about how this policy is structured and implemented in real-world terms. For a better understanding, several aspects of the reform could have been elaborated further, such as:

- Policy Mechanism: While the study mentions that the reimbursement for cross-regional medical services was initially focused on hospitalization and later expanded to outpatient services, it doesn’t explain the process in detail. How do individuals access care in different regions, and what steps are involved in the reimbursement process? How do patients know if they are eligible for reimbursement or how to apply for it?

- Pilot Areas: The authors refer to the pilot regions where the reimbursement system was first introduced, but do not explain how these geographical areas were chosen. Do these regions have specific geographical or demographic characteristics? Are they more urban or rural? Did these characteristics play a role in the selection process?

- Coordination with Hierarchical Medical System: The authors argue that the policy conflicts with the goals of the hierarchical medical system, but they do not fully explain how the hierarchical system is supposed to work in the first place. A clearer explanation of this system, and why the cross-regional reimbursement reform disrupts it, would help readers better understand the policy's implications (see also comment below).

3. Response: We sincerely thank you for your valuable comments. Your thoughtful comments have provided clear guidance on areas of the manuscript that require further improvement. We have made modifications and additions to the introduction, mainly focusing on the policy mechanisms, the selection of pilot regions, and the coordination with the hierarchical medical system. The detailed revisions are as follows:

1) Policy Mechanism:

We have supplemented the revised manuscript with relevant content, systematically reviewing the reform trajectory of the cross-regional medical treatment policy from hospitalization to outpatient reimbursement, and providing a detailed explanation of the specific process for outpatient reimbursement. We have provided a detailed explanation of the policy development process in the first paragraph of the introduction: “As early as March 2009, the Opinions of the Central Committee of the Communist Party of China and the State Council on Deepening the Reform of the Medical and Healthcare System [4] proposed prioritizing the transfer and continuation of basic medical insurance for rural migrant workers and improving settlement services for retirees receiving medical care cross-regional. This initiative launched direct settlement services for cross-regional medical expenses, initially focusing on hospitalization costs [5]. Following the establishment of the National Healthcare Security Administration (NHSA), the scope of cross-regional medical reform has been further expanded. In May 2018, the NHSA launched cross-regional direct settlement for outpatient expenses [6]. By September of the same year, the Yangtze River Delta region (Shanghai, Jiangsu, Zhejiang, and Anhui) was selected as the first pilot region due to its high population mobility and pressing demand for cross-regional medical treatment [7]. In December 2019, the southwest region (Sichuan, Chongqing, Guizhou, Yunnan, and Xizang) and the Beijing-Tianjin-Hebei region also initiated pilot programs for outpatient expense settlement [8]. These pilots were designed to test the policy’s feasibility in regions with varying economic development levels and healthcare resource distributions. On the basis of the initial success of these pilots, in September 2020, the NHSA and the Ministry of Finance jointly issued the Notice on the Issuance of the List of Newly Added Pilot Provinces for Cross-Regional Direct Settlement of Outpatient Expenses, signaling the nationwide implementation of cross-regional direct settlement for outpatient expenses [9].” (on lines 44-60 of the manuscript and lines 51-67 of the revised manuscript with track changes.) We have provided a detailed explanation of the reimbursement conditions and process in the second paragraph of the introduction: “Patients seeking medical treatment cross-regional can complete the required pre-treatment filing procedures in advance through various online and offline channels, including the National Medical Insurance Service Platform app, the mini-program on the State Council mobile for cross-regional medical services, or the service windows of insurance agencies. Once the filing is completed, patients are able to access medical services at designated medical institutions using valid credentials such as the electronic medical insurance certificate or social security card. When settling medical expenses, individuals only need to pay the portion they are personally responsible for, without the need to advance payments or go through additional procedures to receive medical insurance reimbursement [10].” (on lines 63-71 of the manuscript and lines 70-78 of the revised manuscript with track changes.)

2) Pilot Areas

We have provided a detailed supplement in the revised manuscript regarding the selection criteria for pilot regions and their characteristics: “By September of the same year, the Yangtze River Delta region (Shanghai, Jiangsu, Zhejiang, and Anhui) was selected as the first pilot region due to its high population mobility and pressing demand for cross-regional medical treatment [7]. In December 2019, the southwest region (Sichuan, Chongqing, Guizhou, Yunnan, and Xizang) and the Beijing-Tianjin-Hebei region also initiated pilot programs for outpatient expense settlement [8]. These pilots were designed to test the policy’s feasibility in regions with varying economic development levels and healthcare resource distributions. On the basis of the initial success of these pilots, in September 2020, the NHSA and the Ministry of Finance jointly issued the Notice on the Issuance of the List of Newly Added Pilot Provinces for Cross-Regional Direct Settlement of Outpatient Expenses, signaling the nationwide implementation of cross-regional direct settlement for outpatient expenses [9].” (on lines 51-60 of the manuscript and lines 58-67 of the revised manuscript with track changes.)

3) Coordination with Hierarchical Medical System

In the third paragraph of the introduction, we have provided a detailed description of the goals and implementation methods of the hierarchical medical system while also explaining the reasons why outpatient settlement for cross-regional medical treatment impacts this system: “The primary objective of the hierarchical medical system is to optimize the allocation of medical resources by promoting the division of labor and cooperation among various levels of medical institutions. This aims to address the issue of overcrowding in advanced medical institutions and underutilization of resources in primary healthcare fa

---

## [Decision Letter · Decision Letter 1]

2 Apr 2025

The impact of outpatient settlement for cross-regional medical treatment on healthcare choice by the floating population: PSM+DID evidence based on CFPS

PONE-D-24-36231R1

Dear Dr. Sun,

We’re pleased to inform you that your manuscript has been judged scientifically suitable for publication and will be formally accepted for publication once it meets all outstanding technical requirements.

Kind regards,

Matteo Lippi Bruni, PhD

Academic Editor

PLOS ONE

Additional Editor Comments (optional):

Reviewers' comments:

Reviewer's Responses to Questions

**Comments to the Author**

1. If the authors have adequately addressed your comments raised in a previous round of review and you feel that this manuscript is now acceptable for publication, you may indicate that here to bypass the “Comments to the Author” section, enter your conflict of interest statement in the “Confidential to Editor” section, and submit your "Accept" recommendation.

Reviewer #1: All comments have been addressed

Reviewer #2: All comments have been addressed

2. Is the manuscript technically sound, and do the data support the conclusions?

Reviewer #1: Yes

Reviewer #2: Yes

3. Has the statistical analysis been performed appropriately and rigorously? 

Reviewer #1: Yes

Reviewer #2: Yes

4. Have the authors made all data underlying the findings in their manuscript fully available?

Reviewer #1: Yes

Reviewer #2: Yes

5. Is the manuscript presented in an intelligible fashion and written in standard English?

Reviewer #1: Yes

Reviewer #2: Yes

6. Review Comments to the Author

Reviewer #1: All the points raised have been sufficiently addressed and necessary changes have been made to the manuscript. The reference literature has been expanded and the health care reform has been better explained. Finally, the English language has also been appropriately revised.

Reviewer #2: Thanks for considering all my suggestions. In this updated versione the paper is improved and it can be considered for publication.

7. PLOS authors have the option to publish the peer review history of their article (what does this mean? ). If published, this will include your full peer review and any attached files.

**Do you want your identity to be public for this peer review?** For information about this choice, including consent withdrawal, please see our Privacy Policy .

Reviewer #1: No

Reviewer #2: No

---

## [Editor Report · Acceptance letter]

PONE-D-24-36231R1

PLOS ONE

Dear Dr. Sun,

I'm pleased to inform you that your manuscript has been deemed suitable for publication in PLOS ONE. Congratulations! Your manuscript is now being handed over to our production team.

Kind regards,

on behalf of

Dr. Matteo Lippi Bruni

Academic Editor

PLOS ONE